# Artificial light-driven ion pump for photoelectric energy conversion

Kai Xiao [1], Lu Chen[1,2], Ruotian Chen[3], Tobias Heil[1], Saul Daniel Cruz Lemus[1], Fengtao Fan[3], Liping Wen[4,5], Lei Jiang[2,4,5] & Markus Antonietti[1]

Biological light-driven ion pumps move ions against a concentration gradient to create a membrane potential, thus converting sunlight energy directly into an osmotic potential. Here, we describe an artificial light-driven ion pump system in which a carbon nitride nanotube membrane can drive ions thermodynamically uphill against an up to 5000-fold concentration gradient by illumination. The separation of electrons and holes in the membrane under illumination results in a transmembrane potential which is thought to be the foundation for the pumping phenomenon. When used for harvesting solar energy, a sustained open circuit voltage of 550 mV and a current density of 2.4 μA/cm$^2$ can reliably be generated, which can be further scaled up through series and parallel circuits of multiple membranes. The ion transport based photovoltaic system proposed here offers a roadmap for the development of devices by using simple, cheap, and stable polymeric carbon nitride.

[1] Department of Colloid Chemistry, Max Planck Institute of Colloids and Interfaces, 14476 Potsdam, Germany. [2] Key Laboratory of Bio-inspired Smart Interfacial Science and Technology of Ministry of Education, School of Chemistry, Beihang University, Beijing 100191, PR China. [3] State Key Laboratory of Catalysis, 2011-iChEM, Dalian National Laboratory for Clean Energy (DNL), Dalian Institute of Chemical Physic (DICP), Zhongshan Road 457, Dalian 116023, PR China. [4] Key Laboratory of Bio-inspired Materials and Interfacial Science, Technical Institute of Physics and Chemistry, Chinese Academy of Sciences, Beijing 100190, PR China. [5] School of Future Technologies, University of Chinese Academy of Sciences, Beijing 101407, PR China. Correspondence and requests for materials should be addressed to K.X. (email: xiaokai@iccas.ac.cn)

Solar energy conversion in some archaea, such as *Halobacterium halobium*, has much in common with its analogue in green plants but is simpler and better understood[1]. By absorbing energy from sunlight and converting it into electronic excitation energy, these microorganisms can pump protons across the cell membrane, generating an osmotic and charge imbalance which in turn powers the synthesis of adenosine triphosphate (ATP)[2,3]. From a technological perspective, following the generation of an electrochemical gradient is the potential for its conversion to electrical energy in the form of an electric current. The concept of a light-triggered electrochemical gradient and photoelectric conversion are therefore linked[4,5]. Previous work has attempted to realize ion pump function by making artificial photosynthetic centers with intra/intermolecular long-range charge separation[4,6,7], or incorporating a photochromic molecular switch into a membrane to construct photoisomerization-induced proton pumps[8,9]. However, despite all the important advances described in such work[10], synthetic photon-to-gradient conversion is still ineffective when compared with biological light-driven ion pump systems[3], which can pump ions even against a steep concentration gradient to create a membrane potential directly by conversion of light energy. Such light-driven ion pumps, applicable in a wider range of chemical conditions (i.e., solvent, temperature, salinity), would have a myriad of potential applications, but the promising and challenging task of their design has remained open.

Here, we describe a light-induced ion pump system (Fig. 1a) based on a commercial membrane where the cylindrical nanochannels are coated with carbon nitride to form nanotubes. The separation of electrons and holes in carbon nitride nanotube membrane (CNNM) under illumination results in a transmembrane potential which is thought to be at the heart of the pumping phenomenon, because carbon nitride materials, as a well-developed semiconductor and photocatalyst, have strong light absorption[11,12]. In this work, as a proof of concept, we also show that such a set-up can be used for high-performance photoelectric energy conversion based on the active transport of the ion pump. As previously reported, ion transport through charged and confined solid-state nanochannels[13], nanopores[14], or nanotubes[15] under a pressure or a salinity gradient can generate electric potential[16,17] because of selective ions diffusion[18,19], but mainly based on passive transport. For active transport[20–22], however, there is still no artificial system with sufficient performance and enough power to realize photoelectric energy conversion. The simple, cheap, and stable artificial ion pump system described here provides an approach for harvesting solar energy, which may be universal and could work in different salt, acid, and alkali solutions.

## Results

### Fabrication and morphology of the carbon nitride nanotube membrane.
The synthesis used melamine as a starting material in a typical vapor-deposition polymerization (VDP)[23] using an AAO membrane with a pore diameter of 100 nm as a substrate (Supplementary Figure 1). The CNNM inner diameter can be well-controlled from 0 nm (complete filling, i.e., a nanorod) to ~90 nm by changing the amount of melamine (Supplementary Figure 2). For analytical reasons, the carbon nitride nanotube (CNN) can be released by chemical etching AAO substrate with acid[24]. Figure 1b shows a typical optical image, enlarged SEM image of CNNM, and opening view of one single CNN, in which the tube has an external diameter ~90 nm and inner diameter ~30 nm. High-resolution TEM images of a single CNN and electron diffraction show the crystalline pore wall section. The obtained pure nanotubes were thoroughly investigated by X-ray

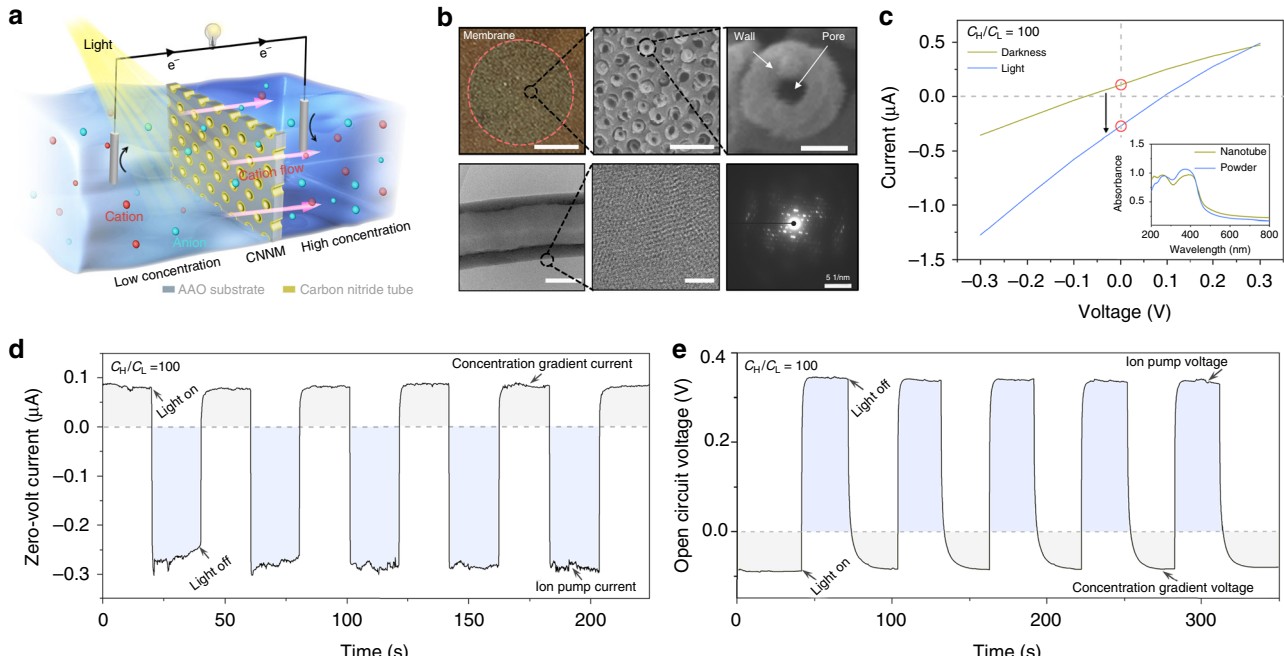

**Fig. 1** Light-induced ion pump based on carbon nitride nanotube membrane. **a** Schematic of the light-induced ion pump, which can pump ions transport against a concentration gradient. **b** Optical image (scale bar 0.2 cm), SEM image (scale bar 500 nm) of CNNM, and typical opening view of nanotube (scale bar 50 nm). The TEM images of a single nanotube (scale bar 50 nm), the enlarged crystalline pore wall section (scale bar 5 nm), and electron diffraction. **c** The typical current–voltage curves before and after light (143 mW/cm²) irradiation at 100-fold ($C_H$ = 0.01 M; $C_L$ = 0.0001 M) KCl concentration gradient. After irradiation, the zero-volt current changed from positive to negative value. The inset shows the light absorbance of carbon nitride nanotube and powder. **d** Measured cyclic constant zero-volt current with the alternating illumination at 100-fold KCl concentration gradient. **e** Measured open-circuit voltage across the CNNM before and after illumination in 100-fold KCl concentration gradient

photoelectron spectroscopy (XPS) and X-ray diffraction (XRD) measurements (Supplementary Figure 3 and Figure 4). All of the results are in excellent accordance with previously reported values for carbon nitride powders (CNPs) with heptazine unit as the repeating unit[11,12]. In addition, the light absorbance of CNNM indicates the potential photo-responsive properties (Fig. 1c, inset).

**Ion pump phenomenon.** The ion pump properties were measured in home-made electrolyte cells (Supplementary Figure 5). The CNNM was symmetrically placed in contact with two KCl solutions differing by a factor of 100 in concentration ($C_H = 0.01$ M; $C_L = 0.0001$ M) and initially illuminated from the low concentration side. Figure 1c shows the typical current–voltage ($I–V$) characteristic of CNNM measured in dark and under simulated solar illumination of 143 mW/cm$^2$ (see Supplementary methods). Without illumination, a positive zero-volt current is generated because of selective ions diffusion driven by concentration gradient[20]. Throughout illumination, the zero-volt current decreases from about 0.1 to −0.3 μA. This change of current indicates the direction of ionic movement is reversed inside the nanotube under illumination, that is, ions are moving against the concentration gradient, which can be further confirmed by inductively coupled plasma (Supplementary Figure 6). By calculation, the changed current translates into that single nanotube can pump ~1500 ions per second against 100-fold concentration gradient, an order of magnitude smaller than bacteriorhodopsin proton pump or halorhodopsin Cl ion pump[3], but an unprecedented breakthrough for artificial ion pump[22]. In addition, the CNNM-based ion pump shows a stable and fully repeatable instant response to illumination (Fig. 1d, Supplementary Figure 7 and Figure 8), which is ascribed to the fast separation and recombination of electrons and holes of CNNM under light irradiation[25]. We also measured the open circuit potential across CNNM before and after illumination for the described 100-fold concentration gradient (Fig. 1e). Before illumination, the transmembrane potential originating from the concentration gradient is about −90 mV, while it jumps to 320 mV after illumination. This concentration gradient- and light-induced potential change is completely repeatable, indicating a strong correlation between illumination and ion pump potential.

Further measurements show that the ion pump "power" is closely connected to the illumination power density. The dependence can be confirmed by the ionic current direction in Fig. 2a. Only at high illumination density (>100 mW/cm$^2$), the CNNM can pump ions transport against 100-fold concentration gradient. To our surprise, the CNNM can pump ions even against

5000-fold concentration gradient with a light illumination of 380 mW/cm$^2$ (Fig. 2b), an efficiency which has not been realized before by artificial ion pumps[20–22] and is comparable to biological ion pump[3]. The CNNM-based pump system also shows an obvious relationship with the light wavelength (Fig. 2c and Supplementary Figure 9). To different monochromatic light with same power density (112.5 mW/cm$^2$), high energy blue light has the strongest "power" to pump ions transport while low energy yellow light almost has no effect. These results are consistent with the light absorbance of CNNM. More importantly, the artificial pump system is universal and makes no difference to different electrolytes including acid, saline, and alkali solutions (Supplementary Figure 10 and Figure 11), which clearly exceeds biological ion pump.

**Mechanism of ion pumping.** The surface charge redistribution of CNNM due to the photo-induced separation of electrons and holes is thought to be the key of the ion pump phenomenon (Supplementary Figure 12). As illustrated in Fig. 3a, the initial CNNM is negative charged because of the incomplete polymerization or condensation with electron-rich –NH terminal group[24]. In this condition, the collected ionic current is positive due to the cation diffusion from the high concentration side to the low concentration side, that is, concentration gradient induced potential difference ($V_{CG}$) across the cation-selective CNNM is the only ionic current source (Fig. 3b)[26]. When illuminated from $C_L$ side, the electrons separated from the holes and moved to bulk carbon nitride or unilluminated side (Fig. 3c)[25], resulting in the positive charged surface in the illuminated side; while the unilluminated side is still negative charged. That is the origin of asymmetric surface charge distribution across CNNM (Fig. 3d). In this condition, the ionic current direction is determined by the net potential ($V_{net} = V_{CG} – V_{CNNM}$), which is the electrostatic potential to accomplish the pumping process (Fig. 3e). The change of surface charge can be confirmed by zeta potential without light irradiation (Fig. 3f) and voltage difference between illuminated side and unilluminated side with light irradiation (Fig. 3g). Before illumination, CNNM is negative charged as a function of electrolyte pH. After illumination, the illuminated side of CNNM changed to positive charged, which can be confirmed by surface photovoltage spectrum (Supplementary Figure 13). The resulted voltage difference before illuminated and non-illuminated sides will drive ions transport against a concentration gradient (Fig. 3g). Such a photo-induced electric field also has been observed in other 2D materials, e.g., graphene membranes[27].

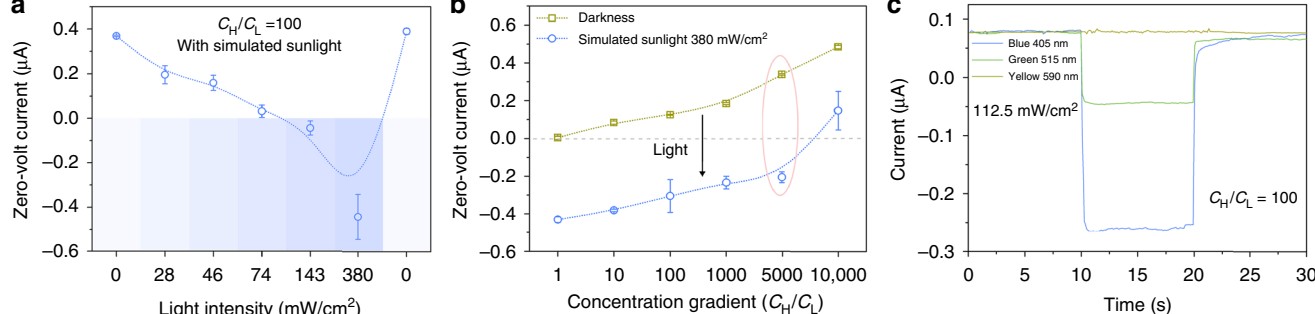

**Fig. 2** High-performance artificial ion pump. **a** Zero-volt current as a function of light density from 0 to 380 mW/cm$^2$. Only the light density stronger than 74 mW/cm$^2$ can change the direction of ionic current at 100-fold KCl concentration gradient. **b** Zero-volt current as a function of concentration gradient from 1 fold to 10,000 fold. The CNNM-based ion pump can realize ions "uphill" transport process at up to 5000-fold concentration gradient. **c** Zero-volt current as a function of monochromatic light (blue: 405 nm; green: 515 nm; yellow: 590 nm) at 100-fold KCl concentration gradient. The ionic current is consistent with the light absorbance. Error bars represent standard deviations from five independent experiments

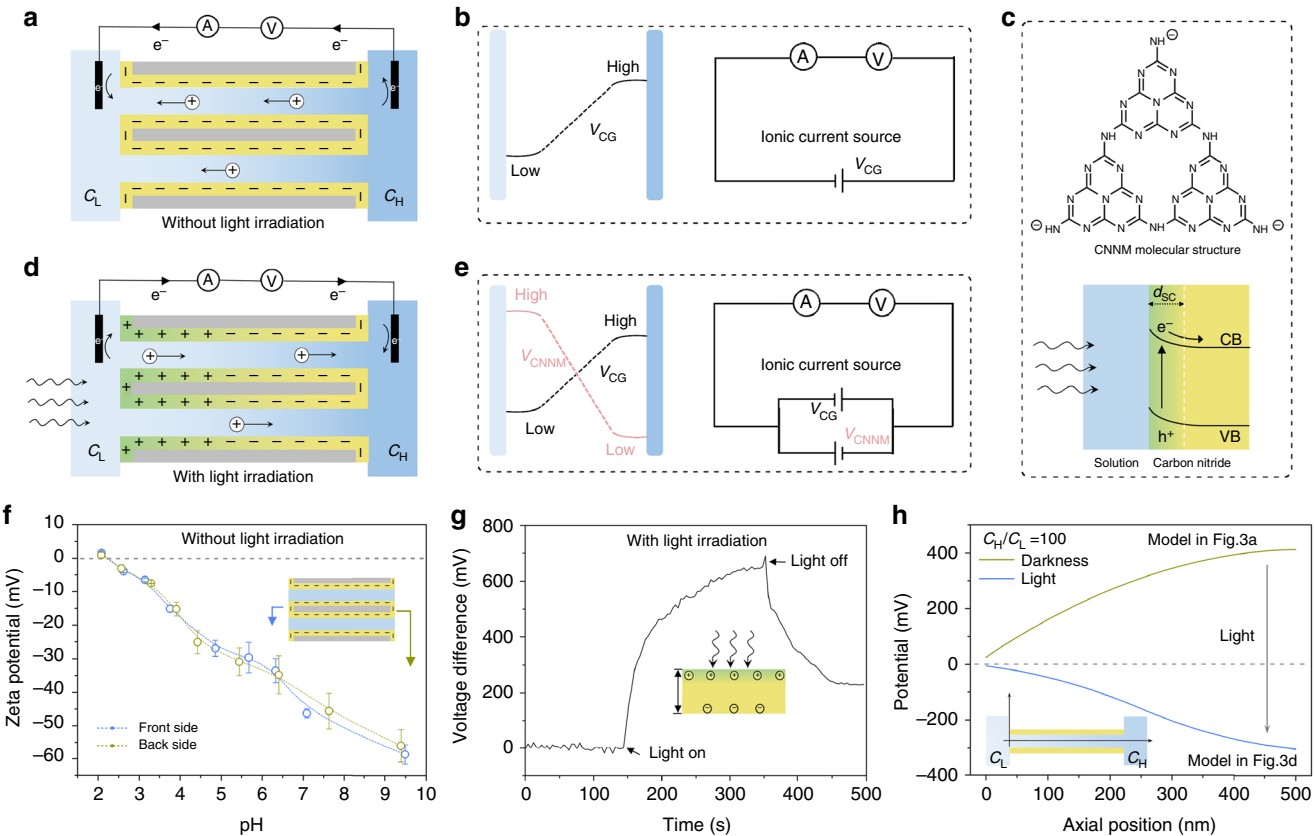

**Fig. 3** Origin of ion pump phenomenon. **a** Schematic of surface charge distribution on the nanotube before illumination, in which condition low-density negative charge is homogeneously distributed on the nanotube. **b** Equivalent circuit of **a**, the concentration gradient potential ($V_{CG}$) is the only ionic current source. **c** Negative charged molecular structure and photo-induced separation of electrons and holes of carbon nitride. **d** Schematic of surface charge distribution on the nanotube after illumination, in which condition the separation of electrons and holes results in the heterogeneous charge distribution. **e** Equivalent circuit of **d**, transmembrane potential ($V_{CNNM}$) provides a contrary potential with $V_{CG}$. **f** Zeta-potential of the CNNM shows the negative charged surface before illumination. **g** The voltage difference between illuminated and non-illuminated sides shows transmembrane potential. Error bars represent standard deviations of three independent experiments. **h** The calculated electrical potential profile along the axis of the nanotube when integrating 100-fold concentration gradient based on the two models illustrated in **a** and **d**, which are based on the 500-nm-long nanotube with 30-nm-diameter. The potential of low concentration side ($C_L$) is preset as 0 mV

To gain more insight into the surface charge redistribution induced potential effect, we further studied the semi-quantitative potential distribution across charged CNNM by theoretical simulation based on Poisson and Nernst–Planck (PNP) equations (Supplementary Figure 14)[28]. To obtain an affordable computation scale, we simplify the CNNM system to a 500-nm-long nanotube (diameter 30 nm) with surface charge distribution shown in Fig. 3a, d. The potential in $C_L$ side was preinstalled to 0 mV. When the CNNM is negative charged with a surface charge density of −0.01 C/m², the potential on the $C_H$ side is about 400 mV higher than $C_L$ side; when the CNNM is asymmetric charged with charge density of 0.01 C/m² in $C_L$ side and −0.01 C/m² in $C_H$ side, the potential in $C_H$ side is about 300 mV lower than $C_L$ side, indicating that the light-induced surface charge distribution can change the voltage direction (Supplementary Figure 15), which is also consistent with the measured transmembrane potential (Fig. 1e).

**Photoelectric energy conversion by the artificial ion pump**. As a proof of concept, this high-performance ion pump also has the potential to be used as an electric generator (Supplementary Figure 16). The CNNM was mounted between two conductivity cells (termed "A side" and "B side") with the same 0.001 M KCl electrolyte (Supplementary Figure 6). A light with a power density

of 380 mW/cm² from A side can produce a stable open circuit voltage up to 550 mV (Fig. 4a). When the photoelectrochemical cell was short-circuited, a photocurrent exhibiting a similar stability of 2.4 μA/cm² was found (Fig. 4b). It is worth mentioning that both the photocurrent and photovoltage are produced and disappeared without hysteresis after the light on and off, which is superior to other ion transport-based energy conversion system[9,29]. The generated power can be output to external circuit to supply an electronic load with output power density up to 1.2 mW/m² (Fig. 4c). It is practically important that the output of CNNM can be further scaled up simply through series and parallel connections of multiple devices. As shown in Fig. 4d, when two CNNMs illuminated by two different light (CNNM 1: 46 mW/cm²; CNNM 2: 143 mW/cm²) separately, they can only generate weak open circuit voltage (CNNM 1: 0.23 V; CNNM 2: 0.42 V) and zero-volt current (CNNM 1: 0.31 μA; CNNM 2: 0.42 μA). When they are connected in series, the photo-induced potential can be scaled up to 0.65 V. When the two CNNMs are connected in parallel, the photo-induced current can be scaled up to 0.71 μA.

Most importantly, the system can be potentially useful for the generation of alternating current by light from different directions. In the light-harvesting system described here, the direction of the ionic current can be controlled by the light irradiation direction, which is superior to other classical

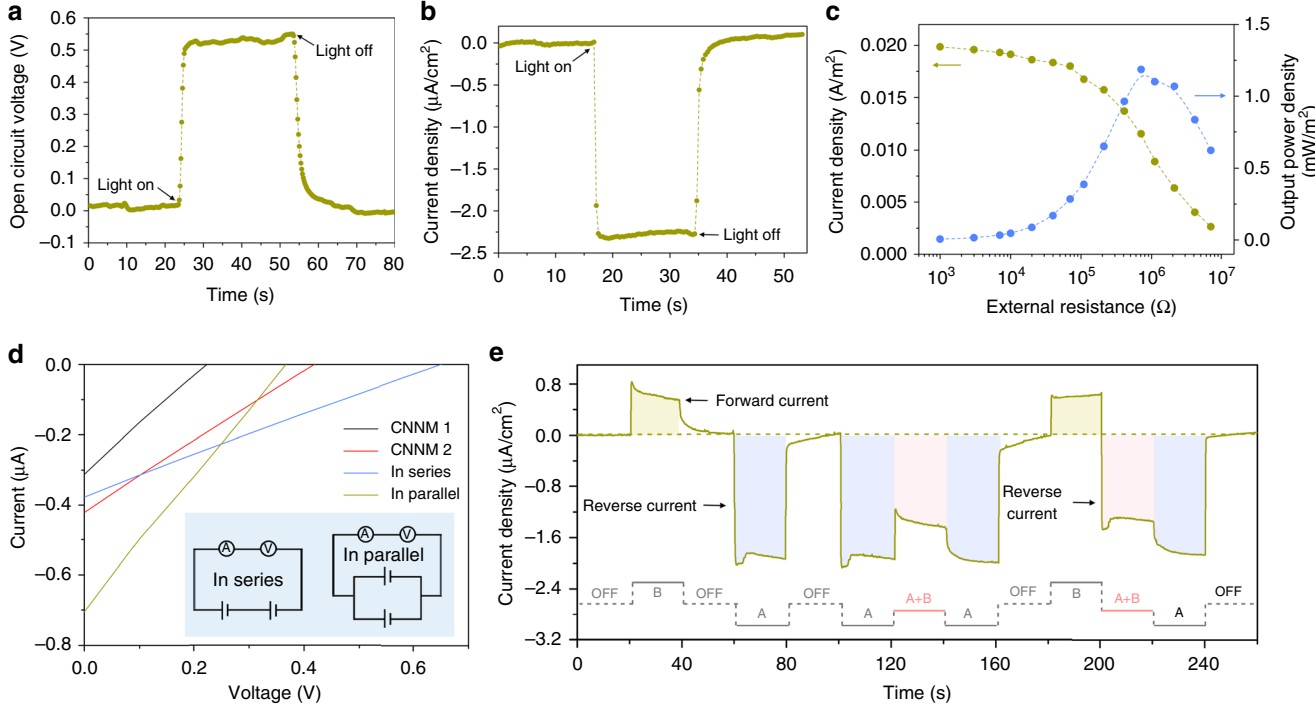

**Fig. 4** Photoelectric energy conversion system based on ion pumping in symmetric electrolyte. The electrolyte is 0.001 M KCl. **a**, **b** Open circuit voltage (**a**) and current density (**b**) generated by light-induced ions transport. **c** The generated power can be output to external circuit and supply an electronic load. The output power density reaches its peak value of 1.2 mW/m$^2$ at resistance of ~400 kΩ. **d** Current–voltage curves of two individual CNNM and their series and parallel connections. Inset: circuit diagram. CNNM 1: 46 mW/cm$^2$; CNNM 2: 143 mW/cm$^2$. **e** Light with different power density (A side: 143 mW/cm$^2$; B side: 46 mW/cm$^2$) from different direction resulted in alternating current

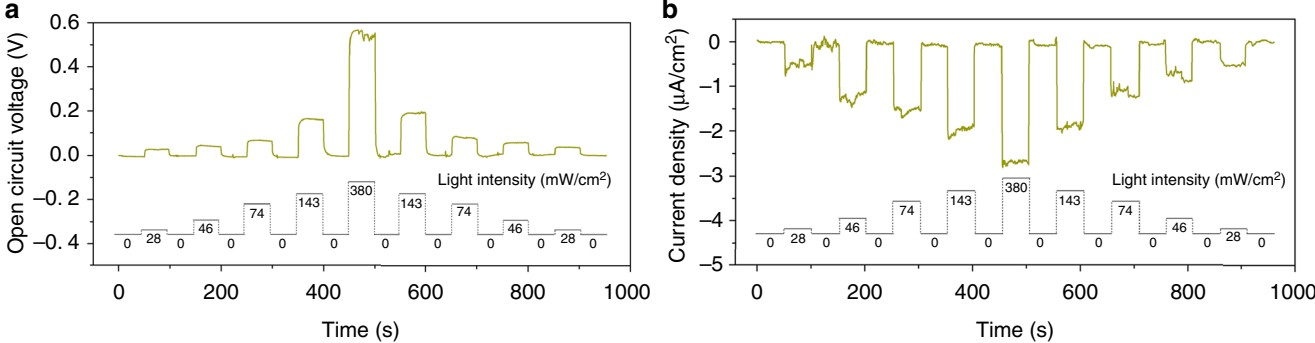

**Fig. 5** Light intensity dependent photovoltage and photocurrent. **a**, **b** Photo-induced voltage (**a**) and ionic current (**b**) as a function of the light intensity from 0 to 380 mW/cm$^2$. Each illumination process continues 50 s

diffusion–osmosis-based system produced by an externally applied pressure gradient[30] and other newly reported electric potential created by, for example, gradient of temperature[31] or chemical concentration[9,26,32]. As shown in Fig. 4e, positive current arising from B side illumination (46 mW/cm$^2$) changes its direction when A side is illuminated instead (143 mW/cm$^2$). Meanwhile, simultaneous illumination on both sides (A + B) generates a mutually offsetting ionic current. In this particular system, different photocurrent patterns can be generated when different light sequences are applied: ion flux follows the light.

The exceptionally high photoresponsivity of our system calls for further characterization. Firstly, both the photocurrent and photovoltage are closely associated with the power density (Fig. 5a, b), a figure of merit for future applications. Secondly, both the photocurrent and photovoltage do not show any sign of deterioration under long continuous illumination (Supplementary Figure 17). Some devices were retested after several weeks

and displayed the same photoelectric energy conversion effect. In principle, the CNNM-based photovoltaic device can be perpetually used for photoelectric energy conversion because the polymeric CNNM is also known in other, more critical experiments to be unchanged before and after illumination[24]. Thirdly, the system also makes no difference to different electrolytes including acid, saline, and alkali solutions for energy conversion, i.e., it is universal (Supplementary Figure 18).

## Discussion

In summary, a polymeric CNN-based photo-induced ion pump system is presented that utilizes the separation of electrons and holes of CNNM under light irradiation. The experimental and theoretical simulation results show that the redistribution of surface charge across CNNM is sufficient to explain the pumping phenomenon. In terms of application in energy conversion, the

photo-induced ion pump system is superior to previously described artificial systems and can compete with and complement biological entities. The ion transport-based photovoltaic system provides an advantageous open circuit voltage exceeding 0.5 V and a pleasing current density under solar-like irradiation. In addition, we may expect performance increases by lowering the length of the nanotubes as well as the diameters to the sizes of protein pumps[3]. In comparison with previously reported methods to harvest light energy[33,34], the approach described here is also cheap, stable, more universal, and efficient.

## Methods

**Fabrication of carbon nitride nanotube membrane**. The CNNM was fabricated by a VDP method described previously[17]. Firstly, the commercial AAO membrane (diameter: 5 mm) were cleaned by ethanol and deionized water, then dried by nitrogen. To control the diameter of CNNM, 0.2, 0.5, 1.0, 1.5, 2.0 g melamine and cleaned AAO were put into the bottom of the glass test tube. The samples were placed in the oven to heat to 773 K with a heating rate of 10 K/min, and then keep for 4 h to insure the sufficient polymerization. After the temperature naturally cooled down to room temperature, the AAO membrane tuned from transparent white to brown, and yellowish CNP at the bottom of the test tube can be obtained (Supplementary Information). To get a pure CNN, the CNNM was immersed in 1 M acid for chemical etching (72 h), then cleaned by deionized water, and dried in 60 °C oven.

**Ion pump and energy conversion properties**. The ion pump and energy conversion properties were studied by measuring the zero-volt ionic current and open circuit voltage through the CNNM with and without light illumination. A CNNM membrane was mounted between two chambers of the self-made cells, which are full of electrolytes with different concentration. To the ion pump properties, low concentration electrolyte was facing the light illumination side, while high concentration electrolyte was facing the darkness side. To the energy conversion properties, the electrolytes in two cells are the same. Ag/AgCl electrodes were used to collect the current and voltage signals (anode facing the $C_H$ side). Open circuit voltage (photovoltage) was measured by electrochemical workstation (Gamry interface 1000). Ionic current (photocurrent) was measured by a Keithley 6430 picoammeter (Keithley Instruments, Cleveland, OH). To the light density-dependent measurements, the light density can be controlled by controlling the distance between light source and CNNM. To realize the series and parallel connections, two homemade devices with CNNMs are connected by wire. A variable resistance was connected to the circuit by wire to measure the output current. The output power densities can be calculated as $P = I^2 \times R_L$, where $I$ is the measured ionic current and $R_L$ is the resistor load.

## Data availability

The data that support the findings of this study are available from the corresponding author upon reasonable request.

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

## Acknowledgements

K. Xiao acknowledges the support of Alexander von Humboldt Foundation. K. Xiao thanks the kindly discussion from Dr. Yubao Zhao, Dr. Ang. Li, Dr. Guigang Zhang, and Prof. Shaowen Cao. This work was financially supported by Max Planck Society and National Key Research.

## Author contributions

K.X. conceived and designed the experiments. K.X. and L.C. performed the research and calculations. T.H. helped measure the TEM. S.D.C.L., R.C. and F.F. helped measure the surface photovoltage spectrum. K.X., L.W., L.J. and M.A. analyzed the data and discussed the results. K.X. and M.A. wrote the manuscript.

## Additional information

**Competing interests:** The authors declare no competing interests.

