## [Peer Review File · Nature Communications]

Reviewers' comments:

Reviewer #1 (Remarks to the Author):

The paper by K. Xiao et al. reports a phenomenon of light-driven transport of ions across carbon-nitride nanotubes (CNNs). The experiments are well performed and the effect is indeed very interesting. The authors shows convincingly that the ionic current across a CNN membrane, generated under a concentration gradient, can be tuned by external light. This has definitely various applications, which the authors indeed discuss.

In my opinion the paper would make a valuable contribution to Nature Communications. However I raise below several important comments on the paper, which preclude publication in the present state and should be definitely be taken into account before acceptance can be reached. This implies important rewriting of parts of the paper.

I have two important issues at this stage with the manuscript:

(1) The wording "ions pump", which is used in the title and all along the paper, is strgonly misleading. Specifically the study reports measurement of an electric (ionic) current under a concentration gradient, whose amplitude and sign can tuned by external light.

On the other hand, the terminology of an ionic pump implies the generation of an ion *flux*, possibly going *against the concentration gradient*. Measuring the current is not a definite proof of "ionic pumping". In particular there is no fundamental principle fixing the sign of the ionic current versus the concentration gradient (it can well be positive or negative). Therefore it is not possible to state, on the basis of the present results, that the reported effect is indeed pumping.

I insist: this does not reduce the importance and relevance of the reported effect. But the terminology of "ionic pumping" is not relevant and should be avoided. The manuscript and its title should be rewritten accordingly.

(2) The interpretation of the effect is also misleading. The authors summarize their interpretation by stating that "redistribution of surface charge across the CNNs [under illumination] is sufficient to explain the pumping phenomenon."

This interpretation is inappropriate and quite dangerous. It implies that a charge distribution along a channel could lead to a current by itself: this is definitely wrong, as it implies perpetual motion. In the presence of a surface charge heterogeneity a Donnan potential builds up to compensate for an inhomogeneity of the ion distribution (associated with a fixed chemical potential at equilibrium). If nothing is consumed, an equilibrium is reached and no current is generated.

This raises the question of the source of the contribution of the light-induced phenomena to the ionic current, which is - and I am not questioning it - indeed measured experimentally. An obvious answer is that the source of energy for the transport is the salinity gradient and the charge redistribution modifies the sign of the electric current (as I mentioned above, no thermodynamic principle does fix the sign of the electric current, the light-induced charge heterogeneity may reverse the current induced by salinity gradient). In this scenario, light acts as a tuning factor for the sign of the current induced by salinity gradient, but is not the source of it.

However, in view of some results, I cannot rule out a supplementary chemical/redox reaction occurring on the illuminated CNNs and triggered by light. In particular, while most reported results are shown in the paper under a non-vanishing salinity gradient ($CH/CL > 1$), I find the result in Fig 1f highly interesting: for $CH/CL=1$, the zero-volt current is measured to be non-vanishing. This suggests that light can act as a driving force for the ionic current per se, without the necessity of an additional salinity gradient.

This mechanism is different from the generation of a surface charge proposed by the authors in the manuscript, since it requires non-equilibrium consumption of chemical species under light.

The authors should report a systematic study of the ionic current versus light intensity for $CH/CL=1$ (similarly to fig4 but salinity conditions are not reported). This would be important in the paper.

Also do the authors have some clue about such possible chemical reactive species at the CNNs ? it would be important to suggest hints about the possible origin of this contribution to the current. Do they observe bubbling on the membrane surface ? Consumption of the CNNs material, eg by comparing CNN before and after light driven currents using SEM images. Etc.

Other comments:

- the manuscript reports value of surface charges used in the simulation in the range: $0.01\text{C}/\text{cm}^2$. This is huge and simply non-realistic. This corresponds to $10^2\text{ C}/\text{m}^2$, which would correspond to 600 charges per nm^2 .

Reviewer #2 (Remarks to the Author):

The manuscript by Xiao, Jiang and Antonietti and co-workers describes an ion pump that is based on carbon nitride nanotubes which are incorporated into an anodic aluminium oxide membrane that separates a high concentration from a low concentration chamber. In this electrochemical cell, if light is shone on the membrane, an electrochemical gradient can be achieved which is very high for an artificial system.

I am no electrochemist and therefore, I cannot comment on the quality of the experiments; it is also possible that I would miss misinterpretations of the data.

However, what I can comment on is the originality of the work. In my view, this is a very smart approach to a very topical problem and the simplicity of the set-up, the fact that the current density is very easily tuneable by the light intensity, the reversibility of the pump commends itself. I like the fact that the authors took the trouble to perform the oxygen reduction experiment and the completeness of the paper almost makes it a full paper.

Therefore, as far as the content is concerned, I would recommend publication unreservedly.

However, what does need to be changed is the language. While the overall structure of the manuscript is good and readable (although I do find Figure 3a a bit small), there are a relatively large amount of grammatical errors that should be addressed by a native speaker. Very often this concerns the use of articles, singular / plural mismatches or sentence structure. I would say, these are "minor" corrections, which are nonetheless very important to give the manuscript the impact it deserves.

Response to Reviewer #1

Comments 1: The paper by K. Xiao et al. reports a phenomenon of light-driven transport of ions across carbon-nitride nanotubes (CNNs). The experiments are well performed and the effect is indeed very interesting. The authors shows convincingly that the ionic current across a CNN membrane, generated under a concentration gradient, can be tuned by external light. This has definitely various applications, which the authors indeed discuss.

In my opinion the paper would make a valuable contribution to Nature Communications. However I raise below several important comments on the paper, which preclude publication in the present state and should be definitely be taken into account before acceptance can be reached. This implies important rewriting of parts of the paper.

I have two important issues at this stage with the manuscript:

(1) The wording "ions pump", which is used in the title and all along the paper, is strgonly misleading. Specifically the study reports measurement of an electric (ionic) current under a concentration gradient, whose amplitude and sign can tuned by external light.

On the other hand, the terminology of an ionic pump implies the generation of an ion *flux*, possibly going *against the concentration gradient*. Measuring the current is not a definite proof of "ionic pumping". In particular there is no fundamental principle fixing the sign of the ionic current versus the concentration gradient (it can well be positive or negative). Therefore it is not possible to state, on the basis of the present results, that the reported effect is indeed pumping.

I insist: this does not reduce the importance and relevance of the reported effect. But the terminology of "ionic pumping" is not relevant and should be avoided. The manuscript and its title should be rewritten accordingly.

Response: Thanks for your comments. We revised the title and the manuscript, but we still think "ions pump" should be left. We accept what you side "Measuring the current is not a definite proof of "ionic pumping"" is right, but only in the case we don't know the surface charge of the nanotube. In fact, ionic current is an effective way to study the ions transport, which has been well studied in the past thirty years (*Science* 1995, 268, 700-702; *Am. J. Phys.*2004, 72, 567; *Nature Chem.*2014, 6, 202-207). By controlling the surface charge of nanochannel/nanopore/nanotube, the direction of ions transport (both cation and anion) is well controlled, and has been confirmed by experiment and calculation (*Adv. Funct. Mater.*2006, 16, 735-746; *Adv. Mater.*2008, 20, 293-297; *Adv. Mater.*2016, 28, 3345-3350). In this work, the surface charge of the nanotube membrane before and after illumination is known, and the reverse of ionic current is origin from the photo-induced redistribution of surface charge (explain later), from which we can know the direction of ions transport (K^+ ions transport against concentration gradient).

To follow the points of the referee and to give clear evidence of the pump process, we directly measured the ions concentration before and after illumination. As show in **Fig. R1a**, electrolytes with same concentration (0.000002 M/L KCl) were put into the cells separated by carbon nitride nanotube membrane. We measured the K^+ ions concentration in B cell before and after light illumination by inductively coupled plasma (ICP) to explain the ions transport. **Figure R1b** showed the K^+ ions concentration of four parallel experiments, which increased obviously after 200 s illumination and can

confirmed the K^+ ions pump process.

Figure R1. (a) Schematic of photo-induced ions pump process. (b) Concentration of K^+ ions measured by ICP before and after pump process (four parallel results). The result clearly indicated that the K^+ ions concentration increased after light illumination.

Comments 2: The interpretation of the effect is also misleading. The authors summarize their interpretation by stating that "redistribution of surface charge across the CNNs [under illumination] is sufficient to explain the pumping phenomenon."

This interpretation is inappropriate and quite dangerous. It implies that a charge distribution along a channel could lead to a current by itself: this is definitely wrong, as it implies perpetual motion. In the presence of a surface charge heterogeneity a Donnan potential builds up to compensate for an inhomogeneity of the ion distribution (associated with a fixed chemical potential at equilibrium). If nothing is consumed, an equilibrium is reached and no current is generated.

This raises the question of the source of the contribution of the light-induced phenomena to the ionic current, which is - and I am not questioning it - indeed measured experimentally. An obvious answer is that the source of energy for the transport is the salinity gradient and the charge redistribution modifies the sign of the electric current (as I mentioned above, no thermodynamic principle does fix the sign of the electric current, the light-induced charge heterogeneity may reverse the current induced by salinity gradient). In this scenario, light acts as a tuning factor for the sign of the current induced by salinity gradient, but is not the source of it.

However, in view of some results, I cannot rule out a supplementary chemical/redox reaction occurring on the illuminated CNNs and triggered by light. In particular, while most reported results are shown in the paper under a non-vanishing salinity gradient ($CH/CL > 1$), I find the result in Fig 1f highly interesting: for $CH/CL=1$, the zero-volt current is measured to be non-vanishing. This suggests that light can act as a driving force for the ionic current per se, without the necessity of an additional salinity gradient.

This mechanism is different from the generation of a surface charge proposed by the authors in the manuscript, since it requires non-equilibrium consumption of chemical species under light.

The authors should report a systematic study of the ionic current versus light intensity for $CH/CL=1$ (similarly to fig4 but salinity conditions are not reported). This would be important in the paper.

Also do the authors have some clue about such possible chemical reactive species at the CNNs ? it

would be important to suggest hints about the possible origin of this contribution to the current. Do they observe bubbling on the membrane surface? Consumption of the CNNs material, eg by comparing CNN before and after light driven currents using SEM images. Etc.

Response: Thanks for your comments and suggestions. Indeed our previous text was phrased either ambiguous if not wrong. The valuable discussion made us to do new complementary experiments and phrase the proposed mechanism more clearly.

The first possible interpretation you proposed is that the source of energy for the transport is the salinity gradient and the charge redistribution modifies the sign of the electric current. We measure the current-voltage curves and ionic current versus light intensity and wavelength for $C_H/C_L=1$ as you suggested. The results stated clearly light-driven ionic current can still be generated and showed a close relationships with light power density and light wavelength (**Fig. R2**), the same with a salinity gradient. Therefore, the salinity gradient is not the source of ionic current.

The second possible interpretation you proposed is that there may be a chemical/redox reaction on the illuminated CNNs and triggered by light. Actually, the reaction in this system can only be a reaction at the electrode. The electrodes we used in the pump system are Ag/AgCl. With light irradiation, K^+ cations were pumped to another side while Cl^- anions still stayed in the illuminated side (**Fig. R3a, left part**). To balance the separation of cations and anions, the remanent Cl^- anions have to undergo a reaction with Ag: $Ag + Cl^- = AgCl + e^-$, while the another electrode has a reaction: $AgCl + e^- = Ag + Cl^-$. By this electrode reaction, the ionic current is transformed into an electronic current (to measure it), and the solar energy was stored in the salinity gradient (**Fig. R3b**). To confirm the mechanism, the Ag/AgCl electrodes were changed to Pt electrode, at which there should no chemical reaction or electrode reaction. K^+ cations and Cl^- anions will accumulate around the Pt electrodes, building up a capacitive energy (**Fig. R3a, right part**). Light illumination charge the system but ionic current decreases gradually because of the generation of the Donnan potential (discussed by the referee). When light was tuned off, the “capacitor” started to discharge with a forward current (**Fig. R3c**). In addition, there are no bubbles and no CNNs consumption, all of these observations confirmed that there is no extra redox reaction in the pumping process.

In addition to that, we also measure the ionic current at different applied bias potentials. **Figure R4** showed that different bias has no obvious influence to the light-induced ionic current, which also confirmed that there is no photocatalysis redox reaction because a bias potential will have a visible influence to photocatalysis (as in photoelectric cells in general).

Figure R2. (a) The typical current-voltage curves before and after light (143 mW/cm²) irradiation without concentration gradient ($C_H=C_L=0.01$ M KCl). (b) The current as a function of light density from 11.8 mW/cm² to 326.5 mW/cm² without concentration gradient ($C_H=C_L=0.01$ M KCl) under 0 V bias. (c) The current as a function of light wavelength without concentration gradient ($C_H=C_L=0.01$ M

KCl) under 0 V bias.

Figure R3. (a) Schematic of ions transport before and after light illumination with Ag/AgCl (left part) and Pt (right part) electrodes respectively. (b) Photo-induced ionic current as a function of the light intensity with Ag/AgCl electrode. (c) Photo-induced charging and discharging process as a function of the light intensity with Pt electrode.

Figure R4. Light-induced ionic current of CNNM at different bias.

Figure R5. (a), Schematic of surface charge distribution on the CNNs before illumination. (b), Photo-induced separation of electrons and holes. (c), Schematic of surface charge distribution on the CNNM after illumination. (d), Zeta-potential of the CNNM shows the negative charged surface before illumination. (e), Surface photo voltage spectrum (SPV) of illuminated side is positive charged. (f), The voltage difference between illuminated side and unilluminated side shows transmembrane potential.

To further confirm the redistribution of surface charge across CNNM, we measure the surface charge before and after light illumination (**Fig. R5**). Before illumination, the CNNM is negative charged because of the incomplete polymerization or condensation with electron rich -NH terminal group, which can be confirmed by the zeta potential (**Fig. R5d**). After illumination, the CNNM changed to positive charged surface in the illuminated side (**Fig. R5e**), resulted in an obvious transmembrane potential (**Fig. R5f**), which can be confirmed by surface photo voltage spectra. In this condition, the ionic current direction is determined by the net potential, which is the electrostatic potential to accomplish the pumping process. In conclusion, the redistribution of surface charge across CNNs should be responsible for the ionic pump, and the solar energy was converted into salinity gradient energy.

Comments 3: - the manuscript reports value of surface charges used in the simulation in the range: $0.01\text{C}/\text{cm}^2$. This is huge and simply non-realistic. This corresponds to $10^2\text{C}/\text{m}^2$, which would correspond to 600 charges per nm^2 .

Response: Thanks for your comments. This is indeed faulty, the surface charge density used in simulation is $0.01\text{C}/\text{m}^2$. This has been corrected in the revised manuscript.

Response to Reviewer #2

Comments 1: The manuscript by Xiao, Jiang and Antonietti and co-workers describes an ion pump that is based on carbon nitride nanotubes which are incorporated into an anodic aluminium oxide membrane that separates a high concentration from a low concentration chamber. In this electrochemical cell, if light is shone on the membrane, an electrochemical gradient can be achieved

which is very high for an artificial system.

I am no electrochemist and therefore, I cannot comment on the quality of the experiments; it is also possible that I would miss misinterpretations of the data.

However, what I can comment on is the originality of the work. In my view, this is a very smart approach to a very topical problem and the simplicity of the set-up, the fact that the current density is very easily tuneable by the light intensity, the reversibility of the pump commends itself. I like the fact that the authors took the trouble to perform the oxygen reduction experiment and the completeness of the paper almost makes it a full paper.

Therefore, as far as the content is concerned, I would recommend publication unreservedly.

However, what does need to be changed is the language. While the overall structure of the manuscript is good and readable (although I do find Figure 3a a bit small), there are a relatively large amount of grammatical errors that should be addressed by a native speaker. Very often this concerns the use of articles, singular / plural mismatches or sentence structure. I would say, these are “minor” corrections, which are nonetheless very important to give the manuscript the impact it deserves.

Response: Thanks for your comments and suggestions. We have revised the manuscript and improved its language based on your suggestions.

REVIEWERS' COMMENTS:

Reviewer #1 (Remarks to the Author):

The authors have answered satisfactorily to my remarks and led complementary experiments which allow to allay all my concerns. This is a nice contribution and I suggest acceptance of the paper.